

# Analysis of fungal dynamic changes in the natural fermentation broth of 'Hongyang' kiwifruit

Zhiming Zhang[1,*], Yuhong Gao[1,*], Wenjuan Zhao[2], Zhuo Wei[2], Xiaozhen Liu[2] and Hanyao Zhang[1]

[1] Key Laboratory for Forest Resources Conservation and Utilization in the Southwest Mountains of China, Ministry of Education, Southwest Forestry University, Kunming, Yunnan, China
[2] Key Laboratory of Biodiversity Conservation in Southwest China, National Forest and Glassland Administration, Southwest Forestry University, Kunming, Yunnan, China
[*] These authors contributed equally to this work.

Corresponding authors
Xiaozhen Liu, 250588726@qq.com
Hanyao Zhang, zhang-hanyao@hotmail.com

## ABSTRACT

'Hongyang' kiwifruit (*Actinidia chinensis* Planch.) is an ideal kiwifruit wine variety. At present, there is no research on the dynamic changes of yeast during the natural fermentation of kiwifruit wine. In this study, a high-throughput was employed to analyze the fungal population composition and diversity in the samples cultured in yeast extract peptone dextrose (YPD) medium and enriched in the natural fermentation process of 'Hongyang' kiwifruit at four time points, day one (D1T), day three (D3T), day five (D5T), and day fifteen (D15T). Five hundred and eighty-two operational taxonomic units (OTUs) were obtained from 131 genera and 178 species samples. The diversity analysis results showed that in the early natural fermentation stage, the dominant species was *Aureobasidium pullulans*, and as natural fermentation proceeded, the genus Pichia became the dominant species. *Pichia kluyveri* was an important species at the later stages of natural fermentation. An analysis of the metabolic pathways shows that *P. kluyveri* plays an aromatic-producing role in the natural fermentation of 'Hongyang' kiwifruit. These results could provide a theoretical basis for the studies of kiwifruit fungal diversity and fungal changes during fermentation. The findings could fix a major deficiency in the production of kiwifruit fruit wine, which lacks a specific flavor-producing yeast species or strain.

## INTRODUCTION

'Hongyang' kiwifruit is the first red-fresh variety of *Actinidia chinensis* Planch. It is bred by the Agricultural Bureau of Cangxi County, Guangyuan, Sichuan Province and Sichuan Natural Resources Research Institute, China. It is the consumers' favorite because of its good quality, delicious meat, strong fruit taste, high nutritional value, and certain medicinal value (*Wang, Li & Wu, 1996*). In addition, because of its high sugar and low acid content, good taste, and red pigment, it is an ideal kiwifruit variety for winemaking (*Wheeler et al., 2008*; *Zhu et al., 2013*). In recent years, the research on 'Hongyang' kiwifruit has been

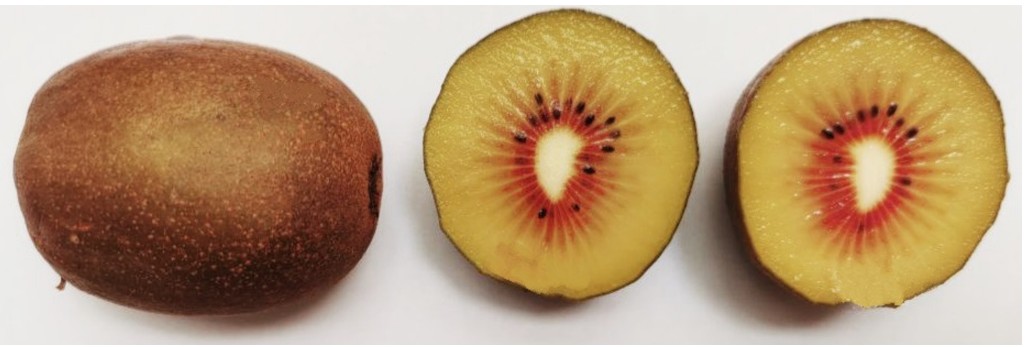

**Figure 1** **The fruit of 'Hongyang' used for experiments.**

focused on breeding, nutritional and medicinal values, and genome analysis. However, no study on the microorganisms attached to 'Hongyang' kiwifruit was reported.

In recent years, the planting area of kiwifruit in China has expanded rapidly. As a result, the output has increased sharply, which has given birth to many new kiwifruit industries, one of which is wine brewing. However, people cannot produce high-quality fruit wine without suitable yeast species or strains. A significant problem restricting the development and production of kiwifruit wine is the lack of specific yeast species or strains, especially those which can produce unique flavors. The yeast usually used in kiwifruit wine production is *Saccharomyces cerevisiae* (*Kang & Zhou, 1992*; *Li, Yao & Zhou, 1999*; *Xia & Zhang, 2000*; *Shang, 2002*). Because it cannot produce special flavors, it cannot meet the needs of kiwifruit winemaking. Therefore, adding new *S. cerevisiae* strains or other yeast species is inevitable to add a flavor. Kiwifruit wine special yeast should be isolated from ripe kiwifruit, and yeast isolated should meet the basic requirements such as aroma production, moderate acid production, and great contribution to wine flavors (*Li, Yao & Zhou, 1999*).

In this study, the dynamic changes of colony structure and metabolic pathway in different periods of naturally fermented 'Hongyang' kiwifruit juice were studied by high-throughput sequencing. The diversity analysis of colony structures and metabolic pathways in different fermentation periods can lay a foundation for the developing by-products of 'Hongyang' kiwifruit. At the same time, it can also provide a theoretical basis for future microbiological research of kiwifruit.

## MATERIALS AND METHODS

### Materials

Fresh 'Hongyang' kiwifruits (see Fig. 1) were collected from Aziying, Panlong District, Kunming, Yunnan Province, China. Sterilized medical rubber gloves were used to pick fresh fruits to avoid the contamination of the samples. The average single fruit of 'Hongyang' kiwifruit weighs 75 g, with red pulp scattered around the central column of the cross-section.

## Yeast in naturally fermented 'Hongyang' kiwifruit juice

Five hundred grams of ripe, fresh, and unsterilized fruits were chopped with the sterilized knife on the ultra-clean workbench. They were evenly placed in three sterile one-liter Erlenmeyer flasks, added 167 mL ddH$_2$O, and fermented at 28 °C. Two milliliters of natural fermentation broth samples on the first day (D1T), the third day (D3T), the fifth day (D5T), and the fifteenth day (D15T), were taken from each Erlenmeyer flask respectively and were placed in a centrifuge tube. Samples were centrifuged in a centrifuge (6000 rpm, 10 min) and washed with phosphate buffer saline (PBS) buffer (1/15 mol/L KH$_2$PO$_4$, and 1/15 mol/L Na$_2$HPO$_4$. 2H$_2$O, pH 7.4) three times. Then cell pellets obtained were stored in liquid nitrogen for high-throughput sequencing.

## Yeast in enrichment culture

After taking and chopping 500g of 'Hongyang' kiwifruit as in the previous paragraph, the fruits chopped were evenly divided into three one-liter Erlenmeyer flasks containing 167 mL YPD liquid medium (yeast extract 10 g/L, peptone 20 g/L, glucose 20 g/L, agar 20 g/L, pH 7.0, 121 °C, sterilization for 15 minutes, stored at 4 °C for later use). Fungi were enriched and cultivated at 28 °C until the liquid medium became turbid, then sampled, and named CK1, CK2, and CK3. Three replicate samples were centrifuged in a centrifuge (6000 rpm, 10 min) and washed with PBS buffer three times, and stored in liquid nitrogen for high-throughput sequencing.

## High-throughput sequencing and bioinformatics analysis
### DNA extraction and sequencing

Total DNA was extracted from each microbial sample using the CTAB method (*Liu et al., 2017*). Hubei Baiqi Biotechnology Co. performed high-throughput sequencing. The sample DNA fragments were paired-end sequenced with the Illumina HiSeq X platform, obtaining the sequences with a mean read length of 150 bases and Q score of 30. The forward primer GGAAGTAAAAGTCGTAACAAGG and the reverse primer GCTGCGTTCTTCATCGATGC were used for the metagenomics sequencing (*White et al., 1990*). The reads were uploaded to GenBank, and the GenBank accession numbers SRR16925436, SRR16945598 to SRR16945606, and SRR16946002 to SRR16946007. Sequencing data were analyzed on the Genescloud platform (www.genescloud.cn).

### Bioinformatics analysis

The sequencing results were obtained by sequence clustering with VSearch (V2.13.4_Linux_x86_64) and MEGA X software to acquire the OTU (Operational Taxonomic Units) representative sequence (*Martin, 2011*; *Rognes et al., 2016*). The representative sequences were compared with the fungal sequences from the Unite database (https://unite.ut.ee/). Species annotation was performed using the default parameters in the QIIME2 software using the pre-trained Naive Bayes classifier (*Kõljalg et al., 2013*). The QIIME feature-table rarefy function in QIIME2 software was used to set the minimum sample sequence size of the leveling depth to 95% to obtain the final OTUs, after removing the singletons.

Krona software (https://github.com/marbl/Krona/wiki) was used to analyze the community taxonomic composition of samples (*Ondov, Bergman & Phillippy, 2011*). R
GGplot2 package was employed to draw a circle stair tree diagram, and the abundance of each out grouping was added to the diagram in the form of a pie chart (*Steenwyk & Rokas, 2021*). To further compare the species composition differences among samples and display the distribution trend of species abundance of each sample, R language and pheatmap package were used to draw heat maps for species composition analysis.

To comprehensively evaluate the alpha diversity of the microbial community, Chao1 and observed species indices were used to represent the richness (*Chao, 1984*), Shannon and Simpson indices for diversity (*Shannon, 1948a*; *Shannon, 1948b*; *Simpson, 1949*), Pielou's evenness index for the evenness, and Good's Coverage index for the coverage (*Pielou, 1966*; *Good, 1953*).

PCoA (Principal Coordinate Analysis) and NMDS (Non-metric Multidimensional Scaling) methods were used to analyze the beta diversity in the sample (*Ramette, 2007*; *Shaffer, Munsch & Juanes, 2018*; *Legendre & Legendre, 1998*). By default, the UPGMA algorithm was used for cluster analysis of the Bray-Curtis distance matrix (*Bray & Curtis, 1957*), and a ggtree of R language was employed to analyze the relationship between different samples for visualization.

Using PICRUSt2 software (https://github.com/picrust/picrust2/wiki), the abundance values of metabolic pathways were obtained. The data generated were placed into the KEGG biological metabolic pathway analysis database (KEGG Pathway Database, http://www.genome.jp/kegg/pathway.html), MetaCyc Database, and metabolic pathways for different sample statistics (*Caspi et al., 2008*; *Langille et al., 2013*). Using R language and MetaGenomeseq package, the Fit Feature Model function was employed, and the distribution of each pathway/group was analyzed by using a zero-log-normal model. The results were used to calculate the significance of the metabolic differences between each natural fermentation sample and CK control group. According to the data selected in the metabolic pathway abundance table, a bar chart was drawn to analyze which species affect the metabolic pathways.

## RESULTS AND ANALYSIS

### Species composition

A total of 1,308,736 sequences were obtained, and the average length of the sequences was 219 bp. According to the 95% similarity, the samples obtained 582 OTUs. In D1T, D3T, D5T, D15T, and CK samples, a total of 131 genera and 178 species were found (See Annex 1). Among the samples from natural fermentation, D1T had 112 genera and 146 species, D3T had 36 genera and 42 species, D5T had 38 genera and 47 species, D15T had 31 genera and 37 species. In the enriched culture samples, CK had 34 genera and 42 species (See Annex 1). As the fermentation of kiwifruit continued, the number of yeast species continued to decrease, and alpha diversity metrics also showed a decreased trend in diversity (see Table 1). In addition, the number of non-*Saccharomyces* species obtained through YPD medium enrichment culture (CK sample) was lower than those of D1T, D3T, and D5T (*p*-Value = 0.000942). See Annex 1 for the relationships between species composition of different samples. The Krona analysis results of all samples are shown in

**Table 1  Alpha diversity indexes among the samples.**

| | Sample | Chao1 | Goods coverage | Observed species | Pielou evenness | Shannon | Simpson |
|---|---|---|---|---|---|---|---|
| | D1T1 | 172.523 | 0.999914 | 171.5 | 0.336702 | 2.49902 | 0.582469 |
| D1 | D1T2 | 135.085 | 0.999979 | 135 | 0.370448 | 2.62159 | 0.675212 |
| | D1T3 | 135.07 | 0.999894 | 133.3 | 0.304578 | 2.14987 | 0.547688 |
| | D3T1 | 33.1276 | 0.99995 | 31.5 | 0.389412 | 1.93791 | 0.672389 |
| D3 | D3T2 | 28.8833 | 0.999964 | 28.3 | 0.390681 | 1.8839 | 0.656108 |
| | D3T3 | 35.7061 | 0.999949 | 34.3 | 0.36583 | 1.86564 | 0.638808 |
| | D5T1 | 44.276 | 0.999929 | 42.7 | 0.372442 | 2.01697 | 0.688565 |
| D5 | D5T2 | 31.02 | 0.999965 | 30.3 | 0.385327 | 1.89599 | 0.66435 |
| | D5T3 | 38.5845 | 0.999929 | 36 | 0.376872 | 1.94827 | 0.671873 |
| | D15T1 | 31.0517 | 0.99995 | 29.3 | 0.319877 | 1.55825 | 0.421895 |
| D15 | D15T2 | 36.0583 | 0.999976 | 35.3 | 0.41005 | 2.10816 | 0.711851 |
| | D15T3 | 37 | 0.999973 | 36.4 | 0.419992 | 2.17781 | 0.610116 |
| | CK1 | 40.9517 | 0.999927 | 38.3 | 0.029358 | 0.154384 | 0.029206 |
| CK | CK2 | 33.631 | 0.999938 | 31.2 | 0.034879 | 0.173098 | 0.035596 |
| | CK3 | 22.4883 | 0.999962 | 21.6 | 0.02647 | 0.117321 | 0.024561 |

Fig. 2. The yeasts accounted for 80% of all the species. Among the yeasts, the *Pichia* genus accounted for the most significant proportion (51%), and it was the dominant genus in the samples (abundance value = 0.409795, *p*-Value = 0.000201). The top ten species in terms of richness for each sample were plotted in a bar chart (see Fig. 3). The dominant fungi in D1T were *A. pullulans* (abundance value = 0.117850, *p*-Value = 0.002684). However, as the fermentation continued, the content of *A. pullulans* and other fungi continued to decrease. On the 5th day of fermentation, *A. pullulans* disappeared. The dominant fungi were *P. terricola* (abundance value = 0.429040, *p*-Value = 0.000021) and *P. kluyveri* (abundance value = 0.392138, *p*-Value = 0.000000). Subsequently, the advantages of *P. terricola* and *P. kluyveri* continued to decrease. On the 15th day, the proportions of *P. terricola* (abundance value = 0.176059, *p*-Value = 0.000021) and *P. kluyveri* (abundance value = 0.176925, *p*-Value = 0.000000) dropped respectively. On the contrary, as the fermentation continued, *P. membranifaciens* (abundance value = 0.573930, *p*-Value = 0.001871) gradually became the dominant species. The amount of *P. kluyveri* showed an increasing trend, followed by a decrease. In the CK, the dominant species were *P. kluyveri* (abundance value = 0.176925, *p*-Value = 0.000000). The highest abundance of *P. kluyveri* was found in the CK, enriched in culture with YPD.

A heat map was drawn for the abundance data of the top 10 species in average abundance to reflect the correlation of fungi among samples and display the distribution trend of species abundance in each sample. The results are shown in Fig. 3. The relationship between each sample and each fungus could be seen from the heat map. The species diversity was the highest on the first day. As the natural fermentation progressed, the diversity gradually decreased. The results of the heat map analysis (see Fig. 4) were consistent with the species composition analysis.

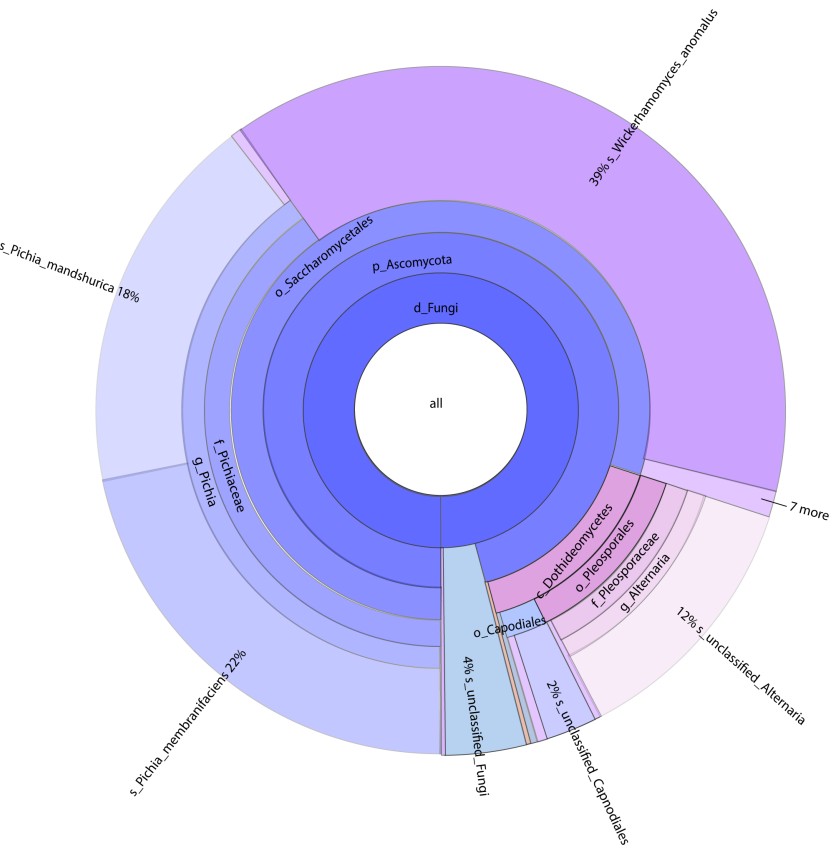

**Figure 2   Classification level and abundance information of Krona diagram of sample species.** From the inside to the outside, the Krona circle represented the seven taxonomic levels of the domain, phylum, class, order, family, genus, and species. The size of the sector reflected the relative abundance of different taxa, and there were specific values. Different colors represented different taxa.

## Alpha diversity analysis

The Alpha diversity index analysis results among the samples are shown in Table 1 and Fig. 5. The Goods coverage values of D1T, D3T, D5T, D15T, and CK in the sample were all about 1 (see Table 1), indicating that the sequencing depth of the sequencing results of all samples this time had covered all species, and the sequencing results represented the yeast in the sample.

The results of the Alpha diversity index analysis revealed that in the natural fermentation process of kiwifruit, the Shannon index of D1T was the highest (2.424493, $p$-Value = 0.000000), and Simpson was relatively low (0.601790, $p$-Value = 0.000003), reflecting the high species diversity of yeast in the sample D1T. The Shannon index value of D15T was the lowest (1.948073, $p$-Value = 0.000000), and the Simpson index value was the highest (0.581287, $p$-Value = 0.000003), reflecting the lowest species diversity in D15T. In the four samples of naturally fermented juice D1T, D3T, D5T, and D15T, the Shannon index first decreased and then increased, and the Simpson index first increased and then decreased, indicating that as the natural fermentation continued, the yeast species diversity

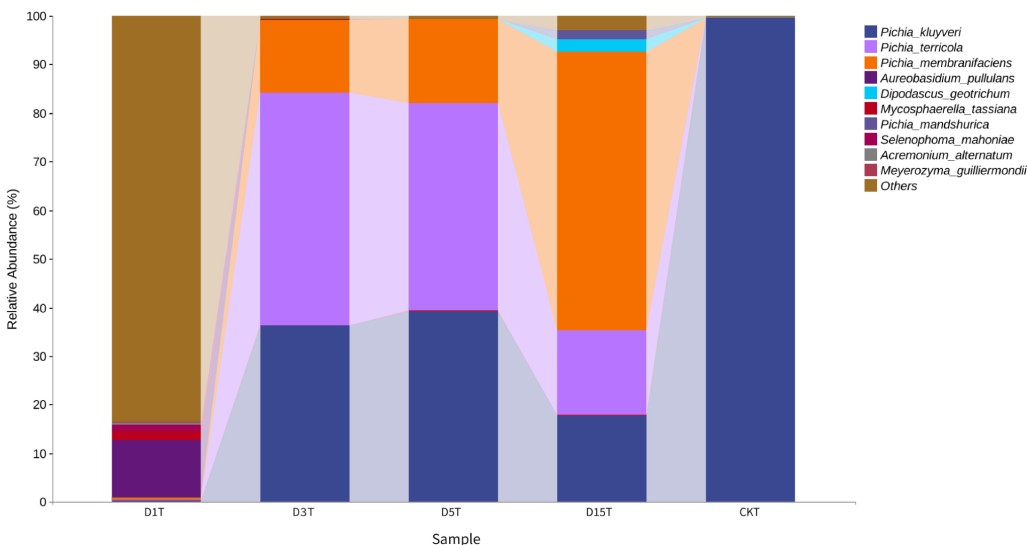

**Figure 3** **Column diagram of horizontal species composition of each sample species.** The abscissa was the name of each group of the grouping scheme, and the ordinate was the relative abundance of each taxon at a specific taxonomic level.

first increased and then decreased. In the enriched culture broth, the Shannon index and Simpson index were smaller than those in the natural fermentation process (Shannon index = 0.148268, Simpson index = 0.029788, and *p*-Value = 0.000000), indicating that the yeast diversity in the kiwifruit enriched culture broth was lower than that in the natural fermentation process.

In the natural fermentation process, the Chao1 index and observed species index of D1T were the highest (Chao1 index = 147.559333, observed species index = 146.600000, and *p*-Value = 0.000001). Among D3T, D5T, and D15T, although the Chao1 index and observed species index of D3T was the lowest (Chao1 index = 32.572333, observed species index = 31.366667, and *p*-Value = 0.000000), indices of D5T was the highest (Chao1 index = 37.960167, observed species index = 36.333333, and *p*-Value = 0.000000), and the overall trend was increasing and then decreasing, but compared to D1T stability means that in the natural fermentation process, the number of species would drop sharply at the beginning of the fermentation and then stabilize. Compared with natural fermentation broth, the Chao1 index and observed species index in the enriched culture broth were lower than those in the natural fermentation broth (Chao1 index = 32.357000, observed species index = 30.366667, and *p*-Value = 0.000000), indicating that the number of yeast species in the enriched culture broth of kiwifruit was lower than the number of yeast species in the natural fermentation broth.

## Beta diversity analysis

The Beta diversity analysis results are shown in Fig. 6. The species composition of the CK samples was far away from the species composition of the natural fermentation samples,

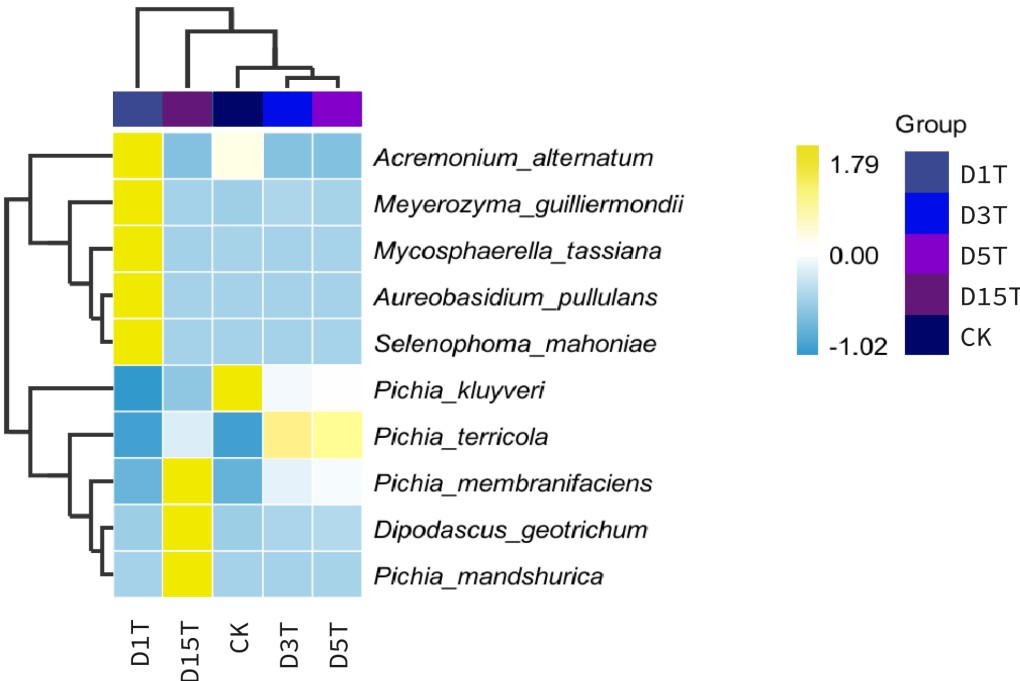

**Figure 4 Horizontal distribution heat map of each sample species.** The samples in the figure were clustered using the UPGMA method, according to the Euclidean distance of species composition data, and were arranged according to the clustering results by default. The species were clustered by UPGMA according to the Pearson correlation coefficient matrix of their composition data and were arranged according to the clustering results.

indicating that there were not many species in the CK samples, and the colony structure of CK samples was quite different from that of the natural fermentation ones.

The similarities among samples were displayed in the form of a hierarchical tree. As shown in Fig. 7, the species composition distance between D3T and D5T samples was the closest ($p$-Value = 0.000147), indicating that the species composition between the two samples was the most similar, and *P. kluyveri* occupied a relatively large proportion (D3T abundance value = 0.363838, D5T abundance value = 0.392138, $p$-Value = 0.000000). On the other hand, the distance between the species composition of the D1T sample and other samples was the furthest ($p$-Value = 0.000080), indicating that the species composition of D1T was different from other samples. The difference between them was enormous. This result was consistent with the results of PCoA and NMDS analysis.

## Functional potential prediction

KEGG database divided natural fermentation samples' metabolic pathways into five categories: biosynthesis, degradation/utilization/assimilation, generation of metabolite and energy, glycan pathways, and metabolic clusters. By comparing the abundance of five kinds of samples, the results are shown in Fig. 8. The high proportion of biosynthesis and precursor metabolite and energy generation indicated that the natural fermentation process was active and produced many by-products. Furthermore, the fatty acid and

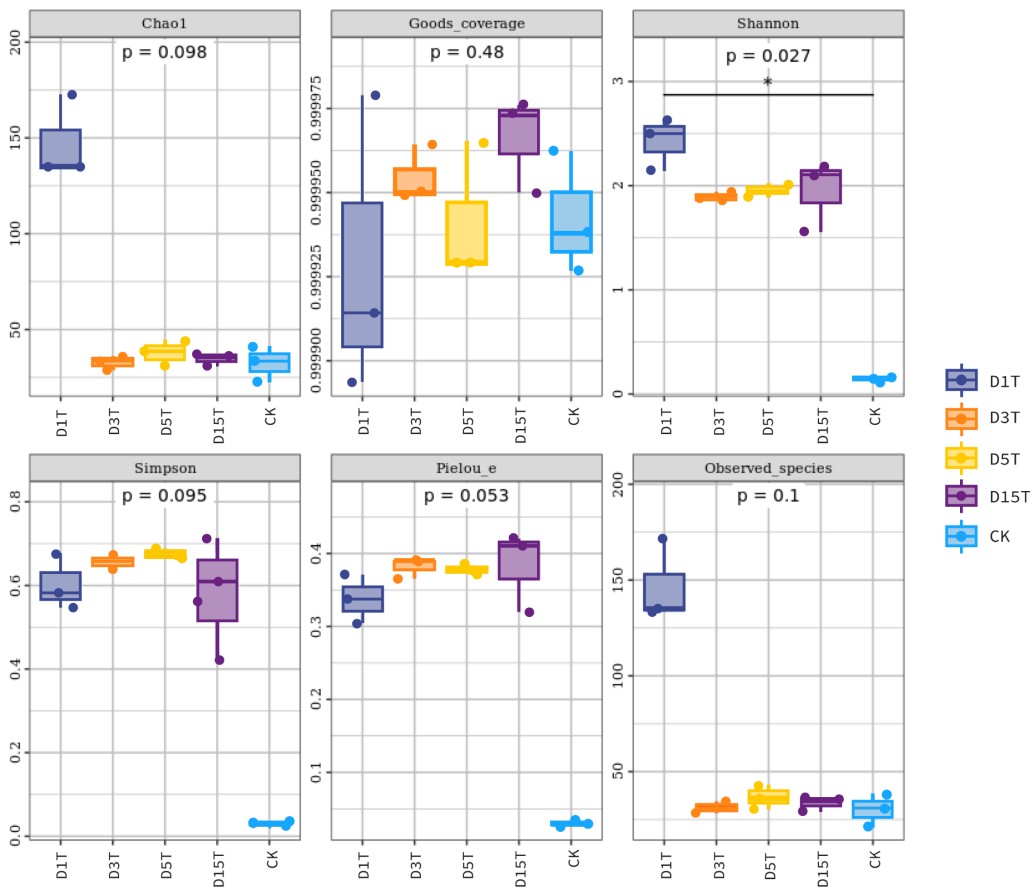

**Figure 5** **Fungal diversity index of samples.** Each panel corresponds to an alpha diversity index, marked in gray at the top. In each panel, the abscissa was the grouping label, and the ordinate was the value of the corresponding alpha diversity index. In the boxplot, the meanings of each symbol were as follows: the upper and lower end lines of the box; the upper and lower Interquartile range (IQR); the median line. Upper and lower edges indicate maximum and minimum values (1.5 times the maximum within the IQR range). The point outside the upper and lower edges indicates outliers.

lipid degradation pathways in the generation of precursor metabolite and energy and degradation/utilization/assimilation were directly related to the production of aromatic ester compounds during fermentation.

To explore the reasons for the changes of metabolic pathways, MetaCyc metabolic pathway was also predicted in this study, took the enriched cultured sample CK as the control group, and analyzed the differences of metabolic pathways in each sample of natural fermentation. Then the metabolic differential pathways following the changes of metabolic pathways in each sample were compared, to clarify which pathways affected the changes of KEGG pathways. The results are shown in Fig. 9.

There is a direct relationship between metabolic pathways and colony structure. It was found that the pathway changes were regulated by *Pichia*, unidentified, *Alternaria*, and unclassified *Ascomycota*; *Pichia* affected most of the pathways (see Figs. S1–S6).
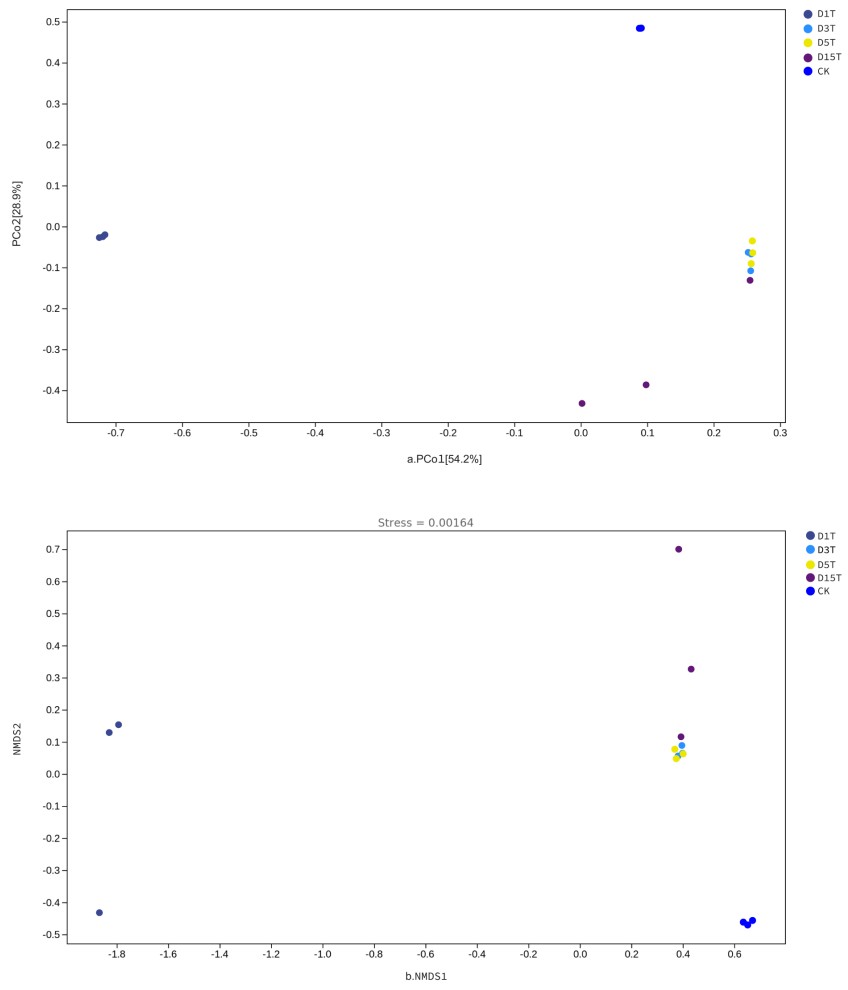

**Figure 6** **The PCoA and NMDS diagram.** (A) PCoA diagram; (B) NMDS diagram. Each dot in the figure represented a sample, and different colored dots indicated different samples (groups).

## DISCUSSION

The traditional isolation technique of plant fruit microbe mainly adopts the method of agar plate diffusion. However, this method has many limitations; it is time-consuming, inefficient, and inaccurate. Therefore, using the traditional analysis method cannot reflect the microbial structure of the fruit very well. With the development of molecular biology, high-throughput sequencing technology has brought good news to the study of microbial diversity with its advantages like high throughput, high sensitivity, high accuracy, and low cost. At present, this technology has been widely used in the studies of microbial diversity (*Reuter, Spacek & Snyder, 2015*). A high-throughput was employed to study the diversities of soil microorganisms during continuous cropping of American ginseng and found that the diversities of soil microbes in continuous cropping of American ginseng decreased, and the domain yeast abundances changed as cropping of American ginseng continued (*Dong et al., 2017*). Furthermore, a high-throughput was conducted on dairy products in Benin,

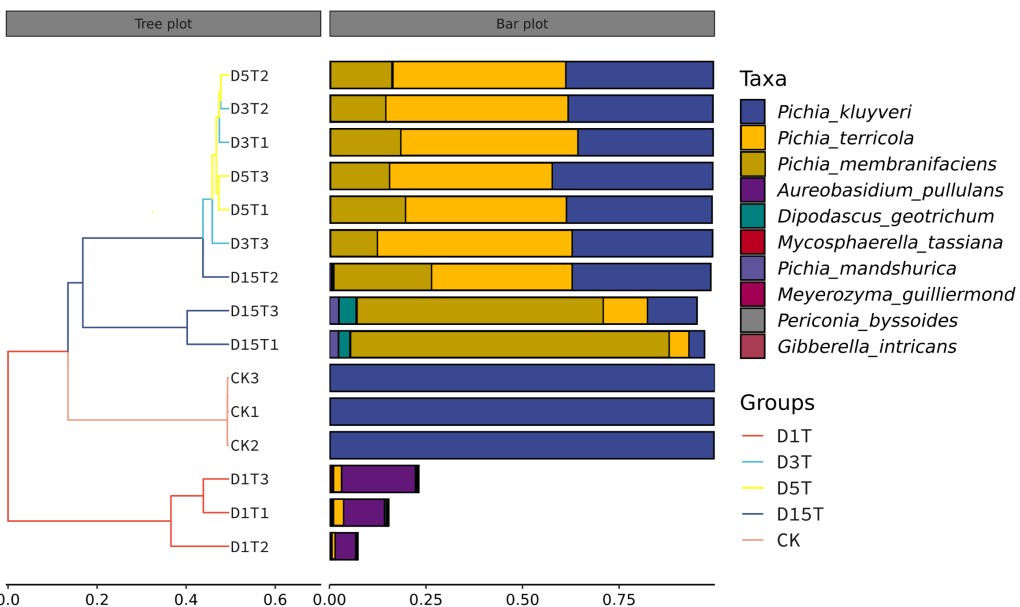

**Figure 7** **Hierarchical cluster analysis among samples.** The panel on the left was a hierarchical clustering tree diagram, in which the samples were clustered according to the similarity between each other. The shorter the branch length between the samples, the more similar the ones were. The panel on the right was a stacked histogram of the top 10 species in abundance.

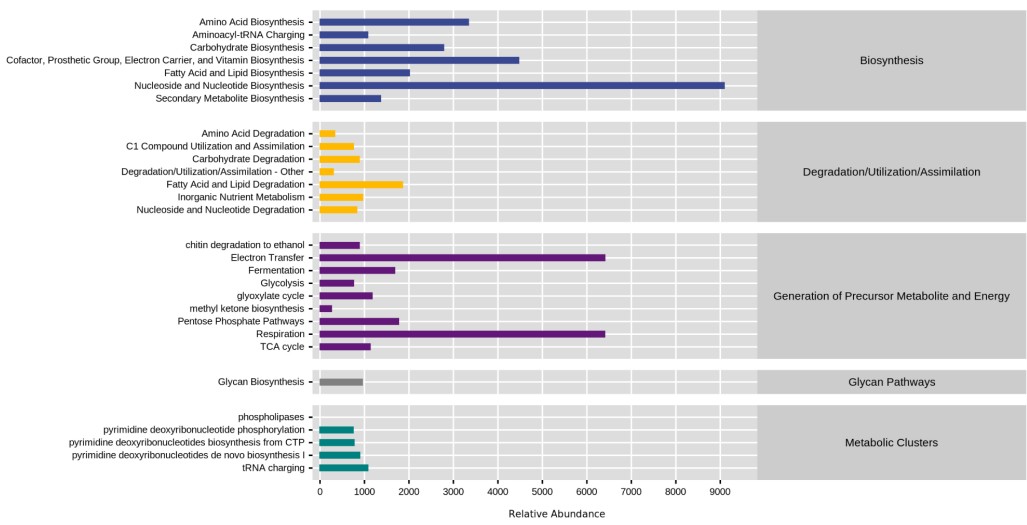

**Figure 8** **Comparison of KEGG first-level metabolic pathways.** The ordinate was the average value of the abundance of functional pathways in the selected samples (in KO/million), and the abscissa was the functional pathways in the first level classification of KEGG. Different colors represented different samples.

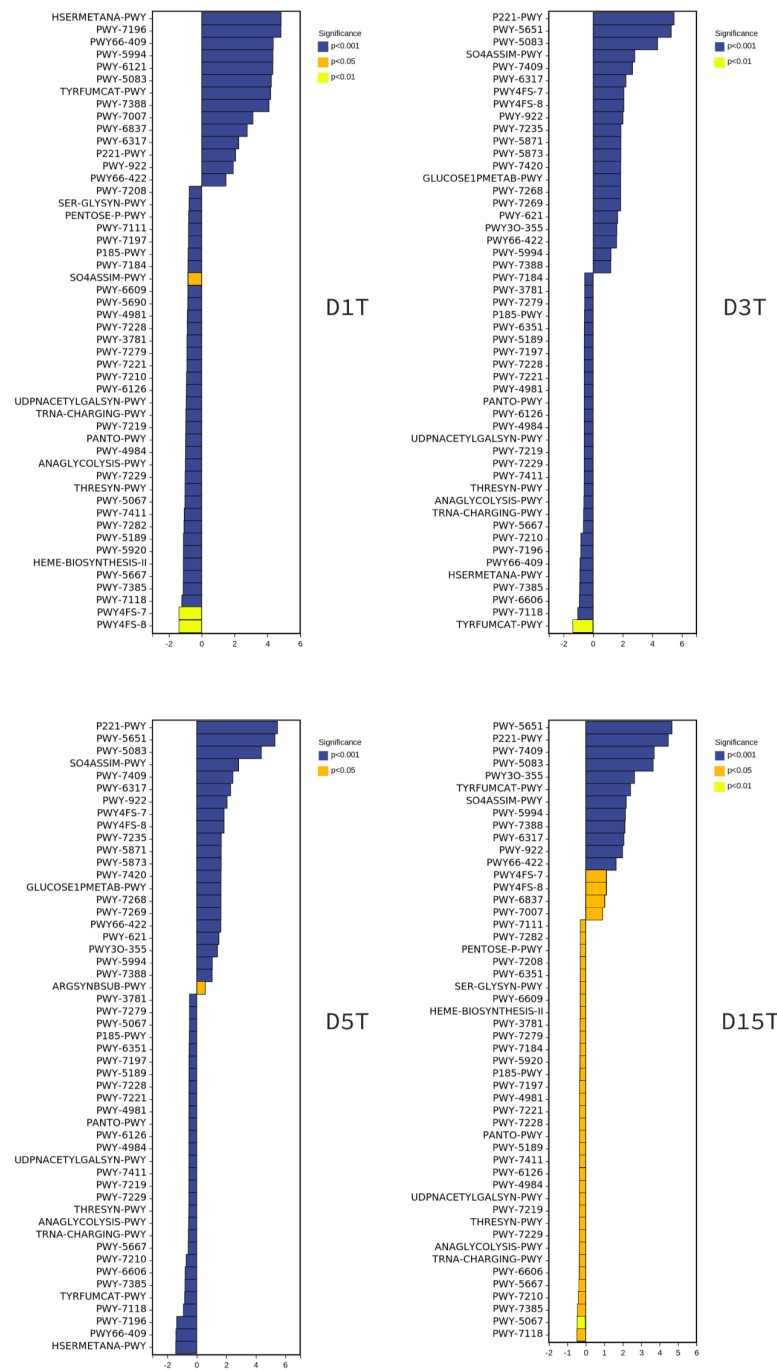

**Figure 9  The metabolic differential pathways with CK as control.** The positive values of horizontal logFC in the up-regulated group were up-regulated relative to the control group, and the negative values were down-regulated. The vertical coordinates were different pathways. Different colors show degrees of significance. The pathway of D1T was significantly different from that of other samples.

Niger, and other places, and found that these dairy products have potential pathogenic yeast *Diatomaceae* (*Sessou et al., 2019*). In this study, we used a high-throughput to study the fungal dynamic diversity of the natural fermentation broth of 'Hongyang' kiwifruit. The results of high-throughput sequencing analysis in this study showed that the dominant yeast in 'Hongyang' kiwifruit was *Aureobasidium pullulans* in the early stage of fermentation (abundance value = 0.117850, $p$-Value = 0.002684). Still, as the fermentation processed, *A. pullulans* gradually disappeared, *P. terricola, P. kluyveri,* and *P. membranifaciens* gradually became the dominant species. In the D15T samples, the amount of *P. membranifaciens* was the largest (abundance value = 0.169346, $p$-value = 0.001871). It indicates that *A. pullulans* is less tolerant of fermentation hyperosmotic environments, and *P. terricola, P. kluyveri,* and *P. membranifaciens* are more tolerant.

In this study, *P. kluyveri* became the dominant species after the third day of natural fermentation (abundance value = 0.363838, $p$-Value = 0.000147) and after enrichment with YPD medium (abundance value = 0.995020, $p$-Value = 0.000000). *P. kluyveri* is one of the most studied non-enological yeasts in fruit winemaking and is considered to be a good fermentation partner of *S. cerevisiae* in fermentation due to its tolerance to the hypertonic environment of the fermentation process and its ability to produce small amounts of ethanol (*Benito et al., 2015*). It has been shown that *P. kluyveri* can produce more 3-sulphonyl hexane-1-ol acetate (3-SHA) compared to other non-winemaking yeasts, and 3-SHA can produce specific aromas. It can increase fermentation esterase activity, catalyze ester formation and produce ethyl 2-phenylacetate, which will produce aromas such as rose or floral notes. Further, it can improve the composition of aromatic compounds (such as thiols, terpenes, fruit esters, etc.), which can produce specific aromas and help improve the overall aroma of fruit wines (*Anfang, Brajkovich & Goddard, 2009*; *Escribano et al., 2017*; *Whitener et al., 2017*; *Ruiz et al., 2019*; *Vicente et al., 2021*). In response to market demand, aromatic compounds are widely used in the pharmaceutical, cosmetic, and food industries (*Fadel, Mahmoud & Asker, 2015*; *Sá et al., 2017*). In the food industry, however, where various chemical additives may be detrimental to human health, yeasts capable of producing aromatic odors through biological activity are preferred to modify food flavors. A previous study concluded that better aromatic qualities were associated with higher fruit characteristics, mainly of peaches, apricots, citrus, and grapefruit, which may be related to the release of thiols during alcoholic fermentation (mainly 3-SHA) increase during alcohol fermentation (*Anfang, Brajkovich & Goddard, 2009*). The flavors of kiwifruit wine have not been investigated, and the presence of large amounts of *P. kluyveri* after natural fermentation in this study may have produced aromatic esters with a special flavor of kiwifruit wine. This study may have solved the problem of the lack of specialized yeast species of kiwifruit fruit wine that can produce specific flavors.

*P. kluyveri* effectively reduces the amount of higher alcohols produced during fermentation (*Benito et al., 2015*). Higher alcohols can mask the flavor of different wines. According to the latest technology in wine production, it is vital to produce wines with as few higher alcohols as possible to keep them below 350 mg/L. The level of higher alcohols after fermentation with *P. kluyveri* and *S. cerevisiae* varies between 176-254 mg/L and below 350 mg/L (*Whitener et al., 2017*; *Ruiz et al., 2019*). The fruit wine industry is

moving towards lower alcohol concentrations, and *P. kluyveri* is a boon to the industry's product upgrade. The addition of appropriate *P. kluyveri* can reduce the ethanol content and improve the specific flavor, increasing the competitive advantage of kiwifruit wine in the industry.

In this study, no *S. cerevisiae* was found in the broth, and may be the reason why alcohol-*active* dry *yeast* should be added to the fermentation broth of kiwifruit. Strains of *S. cerevisiae* for kiwifruit brewing may need to be found in other ways. It has been proved that non-*Saccharomyces* yeast plays a vital and positive role in fruit wine fermentation (*Jolly, Varela & Pretorius, 2014*; *Petruzzi et al., 2017*). Studies have shown that the use of non-yeast yeast during fermentation can effectively reduce ethanol content, increase glycerol content and antioxidant capacity, enrich aromatic components, and produce polysaccharides and mannoproteins (*Benito, 2018*; *Mecca et al., 2020*; *Liu, Lu & Liu, 2021*). Hence, the presence of many non-*Saccharomyces* yeast species in this study also has some value. In this study, we have tried to assess the natural diversity to isolate something that can improve fermentation. However, only *Pichia* was found, which is well known and referenced and present in fruit fermentations but not yet in kiwifruit fermentation (*Vicente et al., 2021*). It is probably why kiwifruit wine is not popular among wine brewers, because it is difficult to find another suitable yeast species or strain producing a distinctive aroma for winemaking from kiwifruits. This is the first paper investigating valuable yeast species or strains for kiwifruit fermentation. In the future, we will carry out additional analyses of the broth during fermentation to determine the reasons for the decline in diversity and seek more beneficial yeast species that can persist and produce specific aromas during the later stages of kiwi wine fermentation.

## CONCLUSION

(1) High-throughput sequencing of the microbial samples from the fermentation broth of 'Hongyang' kiwifruit revealed the presence of many unidentified species, suggesting that new species may be attached to the epidermis of 'Hongyang' kiwifruit. However, the number of unclassified or unidentified species decreased as fermentation proceeded since these fungi could not tolerate the hypertonic environment of natural fermentation broth and were sensitive to environmental changes.

(2) The dominant species in the natural fermentation broth of 'Hongyang' Kiwifruit was dynamically changed as fermentation continued. *P. kluyveri* was the dominant species after the third day of natural fermentation and after enrichment with YPD medium. Based on the analysis results of the KEGG and MetaCyc metabolic pathways, *P. kluyveri* can be applied to the study of kiwifruit wine production and specific flavors. The YPD medium was effective in enriching *P. kluyveri* for use in the kiwifruit wine industry. This research has provided a scientific basis for solving the problem of the lack of particular yeast species and strains of kiwifruit wine that can produce unique flavors.

### Funding
The study was supported by grants from the National Natural Science Foundation of China (32160556, 31760450) and the Joint Project of Agricultural Basic Research in Yunnan Province (2018FG001-038). The funders had no role in study design, data collection and analysis, decision to publish, or preparation of the manuscript.

### Grant Disclosures
The following grant information was disclosed by the authors:
The National Natural Science Foundation of China: 32160556, 31760450.
The Joint Project of Agricultural Basic Research in Yunnan Province: 2018FG001-038.

### Competing Interests
The authors declare there are no competing interests.

### Author Contributions
- Zhiming Zhang performed the experiments, analyzed the data, prepared figures and/or tables, authored or reviewed drafts of the paper, and approved the final draft.
- Yuhong Gao analyzed the data, prepared figures and/or tables, and approved the final draft.
- Wenjuan Zhao performed the experiments, authored or reviewed drafts of the paper, and approved the final draft.
- Zhuo Wei analyzed the data, prepared figures and/or tables, and approved the final draft.
- Xiaozhen Liu and Hanyao Zhang conceived and designed the experiments, authored or reviewed drafts of the paper, and approved the final draft.

### Data Availability
The sequences described here are available at GenBank: SRR16925436, SRR16945598 to SRR16945606, and SRR16946002 to SRR16946007 and at figshare: Zhang, Hanyao (2021): Analysis of Fungi Diversity in the Natural Fermentation of 'Hongyang' Kiwifruit. figshare. Dataset. https://doi.org/10.6084/m9.figshare.17012555.v1.

### Supplemental Information
Supplemental information for this article can be found online at http://dx.doi.org/10.7717/peerj.13286#supplemental-information.

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
