# Peer review of "Analysis of fungal dynamic changes in the natural fermentation broth of ‘Hongyang’ kiwifruit"

_PeerJ, doi:10.7717/peerj.13286_

## Round 0.1 · original submission · Major Revisions

Please rework your manuscript according to the requests from our three reviewers. You must thoroughly address especially the critical comments by reviewer #3

Reviewer 1 ·

Basic reporting

English must be revised.

Experimental design

The authors use a medium-enrichment and high throughput sequencing method to research the fungal communities during the natural fermentation process. However, fungal communities in wine and medium are different because of the inhibitory effects of acid and alcohol.

Validity of the findings

no comment.

Annotated reviews are not available for download in order to protect the identity of reviewers who chose to remain anonymous.

Reviewer 2 ·

Basic reporting

This manuscript studied the fungal changes in the natural fermentation broth of “Hongyang” kiwifruit. Overall, the manuscript should be well organized and written should be improved.

Experimental design

The experimental design was good. Detailed methods should be provided

Validity of the findings

The findings in this manuscript were good. But the results should be clearly demonstrated and improved.
1)The abstract needs to be simplified
2)L37-L49, lots of duplicate content, needs to be clear and precise.
3) Is it necessary to have so many citations? L51, L53, L55
4) L61-67. Please reorganize the sentences.
5) 2.1 Materials. Please provide more detailed information about the kiwifruit used in this study. Such as weight, color, it’s better to have image in the paper. Also, please detail the method and group assignment in the text.
6) Why the wash buffer different? L78-79 vs L87
7) Please add P value throughout the text.
8) L172, what does “the others” mean?
9) There are a lot of description “the most dominant were others”, if others indicated one fungus or yeast, please specify, if others mean a cluster of fungi or yeast, then it cannot be considered as the most dominant. This is important and should be clarify in the text.
10) Please check all the citations, the format should be the same.
11) Some description in the discussion should be moved to the introduction section. And the discussion should be improved.
12) Why only chose “Hongyang” kiwifruit, do you have some control samples in your experiment. It’s good to choose some regular sample as control so that we know it worth it to study “Hongyang” kiwifruit.

Additional comments

Please concise the manuscript.

Annotated reviews are not available for download in order to protect the identity of reviewers who chose to remain anonymous.

·

Basic reporting

Overall the English language used is not bad, but there are some instances where the language has a “google translate-y” feel to it, and needs improvement to get it to an international standard. I suggest you have a colleague who is proficient in English and familiar with the subject matter review your manuscript or contact a professional editing service.

Some examples (there are many more) where the language could be improved include:

Line 12 – has high sugar content -> change to: has a high sugar content
Line 15 – the high-throughput sequencing method -> a high-throughput
Line 17 – in four stages such as one… -> change to: at four time points, day one (D1T), day three (D3T), day five (D5t) and day fifteen (D15T).
Line 23 – metabolic pathway -> change to metabolic pathways
Line 59 to 60 - ..which can understand the change process of colony structure and metabolic pathway and monitor these indexes -> I’m unclear what you are trying to say here, please make it more clear.
Line 106 – space between 2.4.3 and Species
Line 156 – the singleton -> change to singletons
There are many more examples…

Line 49 – Please provide literature references for the few studies that have been done.

Line 50 to 57 – I personally feel these lines can be deleted. This now well-known knowledge and does not contribute to the science of the paper.

Line 64 to 67 – Its different than lines 61 to 63, but still feels like repetition, I suggest keeping only one of the two sentences.

Line 154 – Results: Results are to long and there is a lot of non-scientific descriptions of the figures. You don’t have to go “this is high and this is low but its higher than that but lower than this” we can see that on the graphs, you should just be giving the core message of the graphs as the importance will be discussed in the discussion.

Line 295 – Kiwifruit wine special yeast is needed to explore – I’m unsure what this heading is supposed to say, please write it better

Experimental design

The experimental design, how it is reported, aims and scope of the paper is very poor and has to be reworked.

Line 74 – What precautions were taken to not contaminate the samples? Where the outside of the fruit sterilized? Was a sterile surface and knife used?

Line 75 – It’s a bit unclear what was done here. Was the fruit just placed as is in the Erlenmeyer? So was something added? What was then sampled? Some liquid (how much) or pieces of fruit (how much)?

Line 76 – You can remove the word “named” before the sample names

Line 77 – What does “three replicates were set” mean? Did you take replicate samples. How does this fit in with the 500g fruit chopped? Where there multiple Erlenmeyers? Please expand on the experimental setup. How many samples were prepared? It seems like there was just one Erlenmeyer and you took multiple samples from that, which would not be triplicates.

Line 70 – What was then obtained? A cell pellet? Was this stored?

Line 81 – This whole section needs the same attention as I explained with the previous section. Was it just 500g in one Erlenmeyer? What volyme of YPD is in the flask? If its not 500g in one flask, but 500g was chopped and split into various flasks, into how many and how much was placed in each flask?.

Line 86 – I still do not understand what “replicates were set” means.

Line 90 – Section 2.4.1: This section needs a lot of extra info. You can look at the methods of this paper for some guidance: https://doi.org/10.3389/fmicb.2020.01451. You need to include the sequencing platform (e.g. Illumina MiSeq) (I see Line 97 at least says Illumina, but include specific machine), references for the primers, length of sequencing (e.g. 2 x 300) etc. VERY IMPORTANT reads need to be publicly available, so you need to upload them to a database (e.g. SRA) and then provide an accession number.

Line 99 – Cutadapt is for cutting adapters and primers and stuff, so not really for clustering.

Line 102 – change Qiime2 to QIIME2

Line 108 – R is a programming language, GGraph, GGplot, etc, are packages written in R. They also have references that you should include.

Line 112 – Out dots should probably be OTU dots

Line 112 – Qiime2 should be QIIME2

Line 106 to 142 – I recommend consulting literature to write this better. These are standard methods by now and any recent microbiome paper will help you to write it better. Too much information is given and too much repetition is present.

Validity of the findings

Overall the paper is a bit all over the place and can benefit from being more focussed. The title states “Analysis of fungal dynamic changes in the natural fermentation broth of ‘Hongyang’ kiwi fruit ”. This is presented, kiwi fruit is naturally fermented and samples are taken over time and results show a shift in the diversity. BUT then there is this whole second leg to the paper about how high-throughput sequencing will help the commercial fruit wine sector and how this paper has helped to identify yeasts that can benefit the commercial sector and that part of the paper falls flat. This paper should either be diversity changes during natural fermentation, but the literature should then support natural fermentation information and why that is important and then you will have to do additional analysis of the broth during fermentation to determine why the diversity shifts OR this paper should go with the commercial angle, assess the natural diversity in an attempt to isolate something that can improve fermentation and then isolate it, make a fruit wine with and without your isolate and determine if it adds anything. As is, Pichia is well known and referenced and present in fruit fermentations, and this paper as is, just does not add anything new what is already known.

Line 156 – This is a weird way to report the data. Singletons are OTUs consisting of only one read. So you would QC your reads with certain cut-off metrics and report the amount of reads you worked with. Then you would get OTUs during the analysis and filter out singletons and the report the amount of OTUs you have (removing singletons is usually just mentioned in Materials and Methods).

Line 159 – Species composition: I’m not going to go into too much depth here, but your reporting this wrong. During taxonomic assignment QIIME will attempt to assign taxonomy as best it can to a taxonomic level depending on the database, the reads, etc. So some OTUs it will be able to annotate down to species and some only to genus, some only to phylum (it happens), so it did not “find” X amount of genera and Y amount of species. Secondly, if it could annotate down to species, it had to have annotated down to genus before that. Thirdly, its generally accepted that we do not go down to species level with Illumina reads since the reads are so short, but some people do it, I’m personally against it. That is why people would rather do, e.g., PacBio sequencing if they want to go down to species level. If the other reviewers are fine with it, leave it, otherwise you’ll have to change it.

Line 164 – You should use your alpha diversity metrics to show a decrease in diversity.

Additional comments

Overall this paper feels like an attempt to "just publish" the work. I think it can be reworked into something publishable, but as is, I do not feel it is of sufficient quality to justify publication. With a focused aim and scope and results, discussion and conclusion (as well as a small amount of extra lab work) this can be made publishable.

---

## Round 0.2 · accepted · Accept

In spite of the negative opinion of reviewer#1 regarding the novelty of your work, I decided to accept your manuscript due to the paucity of current information on fermentation of kiwifruit wine. I look forward to receiving additional submissions dealing with this interesting subject.

Reviewer 1 ·

Basic reporting

no comment

Experimental design

The authors use a medium-enrichment and high throughput sequencing method to research the fungal communities of kiwifruit as ‘CK’. But I think that's wrong. Fungal communities in medium and kiwifruit are different.

Validity of the findings

The results lack novelty. Pichia is well known in fruit fermentations, and this paper does not add any new findings what has been reported. The research results have no guiding significance for the production of kiwifruit wine.

Annotated reviews are not available for download in order to protect the identity of reviewers who chose to remain anonymous.

Reviewer 2 ·

Basic reporting

Overall, the manuscript was improved after the revision. The authors clarified some comments. As for the figures in the manuscript, all the font size are different, please make sure the figures are in the same format.

Experimental design

good

Validity of the findings

good

·

Basic reporting

Overall English and writing throughout the paper was improved. Literature was added where needed.

Experimental design

Research question much better defined. Streamlining of paper also highlights the core better.

Validity of the findings

Finding are (and were) interesting.

Additional comments

Paper was improved and most comments adequately covered.